# Biodiversity of *Aspergillus* Species and Their Mycotoxin Production Potential in Dry Meat

**DOI:** 10.3390/foods13203221

**Published:** 2024-10-10

**Authors:** Toluwase Adeseye Dada, Theodora Ijeoma Ekwomadu, Lubanza Ngoma, Mulunda Mwanza

**Affiliations:** 1Department of Animal Health, Faculty of Natural and Agricultural Sciences, Mafikeng Campus, North-West University, Private Bag X2046, Mmabatho 2735, Mafikeng, South Africa; theodora.ekwomadu@nwu.ac.za (T.I.E.); lubanza.ngoma@nwu.ac.za (L.N.); 2Department of Agricultural Technology, School of Agriculture, Ekiti State Polytechnic, Isan-Ekiti PMB 1101, Nigeria

**Keywords:** *Aspergillus*, biodiversity, aflatoxins, dried meat, phylogeny

## Abstract

This study aimed to examine fungi diversity in dried beef meat sold in Ekiti State, characterize the isolated fungi, and determine the aflatoxin-producing ability of the *Aspergillus* fungi in the samples. Dried beef meat was collected from different markets in Ekiti State and screened for the presence of filamentous fungi using molecular methods. Samples were cultured aseptically on potato dextrose agar (PDA) for fungi isolation, and molecular identification was performed using DNA extraction, Polymerase chain Reaction (PCR), ITS-1/ITS-4 primer pair, and nucleotide sequencing. The results obtained indicated a range of filamentous fungi genera including *Aspergillus*, *Rhizopus*, *Penicillium*, *Fusarium*, *Cladosporium*, *Alternaria*, and other fungi species contaminating the dried meat at (43%), (42%), (3%), (2%), (2%), (1%), and (7%), respectively. High incidences were recorded for *Aspergillus flavus*, *Aspergillus niger*, and *Aspergillus fumigatus* in most of the screened samples. *Aspergillus flavus* accounted for (24.7%) of all the *Aspergillus* species isolated with the presence of the gene needed for aflatoxin production. The occurrences of these filamentous fungal species pose a cause for concern, as most of these fungal species are known producers of certain toxic substances. Maximum likelihood phylogenetic analysis showed a high similarity index score, which indicated a good relationship between isolated *Aspergillus* Species and the closely related strains from GenBank, isolated from different sources and countries. The implication of this study is that consumer health may be at risk through exposure to contaminated dried meat.

## 1. Introduction

Meat is defined as the flesh of animals used as food, and the main sources of high-quality protein are meat, poultry, fish, eggs, nuts, and legumes and dairy products such as milk, yogurt, and cheese [1,2]. The consumption of meat is increasing worldwide, driven by population growth and rising incomes. It is high in minerals (iron, zinc, and selenium) and vitamins (vitamin D, riboflavin, and B_12_), as well as all of the essential amino acids [3]. Fresh meat is highly perishable owing to its biological structure; its preservation and freshness can be altered by various factors, including temperature, packaging, endogenous enzymes, moisture, light, and microbes [4].

The traditional practice of keeping meat in dried form is popular among Nigeria’s rural population, especially for people who are in their late fifties and health-conscious consumers, and this is due to its low-fat and high mineral content, also partly owing to the lack of functionality and availability of modern storage facilities required to maintain them [5]. Therefore, processing them into a dry state for preservation is a necessity in order to have them available for consumption. Examples of dried meats include but are not limited to Banda (boiled and sundried meat), Balangu (smoked chunks of meat), and Kilishi (sliced and sundried meat), the most well-known products [5]. Dried meat is used as an appetizer, a light refreshment to entertain guests, and as a delicacy with or without beverages, and it can be stored for six-to-twelve months.

These traditional methods make dried beef susceptible to fungal and other food pathogen contamination along the processing, drying, and storage stages [6]. When traditionally processed, dried meat products are typically stored in wooden baskets of varying sizes covered with clean cement paper bags, clothes, polythene bags, jute bags, trays, and cardboard boxes, whereas others are casually piled on polythene on the floor in unventilated rooms. Other factors that promote the growth of fungus on dried meat include the quality of meat in raw-material form, physical and biochemical parameters, processing techniques, and hygiene conditions [7].

The occurrence and development of fungi on dried meats decreases quality and raises safety concerns, resulting in significant monetary loss. Reference [8] observed that the presence of undesirable fungal growth may lead to undesirable economic loss to the producers by increasing production cost and product losses. It might also represent a potential health hazard to consumers if the fungal contaminants are pathogenic or toxigenic. The meat is then rejected by consumers due to discolouration and low quality of the meat product. Several studies have shown the presence of fungi and mycotoxins in kundi, fish, banda, and other dried products. Some of them are shown to have the potential to produce secondary metabolites known as mycotoxins, which are capable of remaining in the product long after the fungi have died. Mycotoxins can lead to a range of harmful health effects and present a significant risk to both humans and livestock. These effects vary from acute poisoning to more serious long-term conditions, such as immune system deficiencies and cancer [9,10].

The consumption of meat contaminated with *Aspergillus* species can expose consumers to aflatoxicosis, a disease caused by the ingestion of aflatoxins. These mycotoxins, mainly produced by *Aspergillus flavus* and *Aspergillus parasiticus*, are among the most toxic, with Aflatoxin B_1_ (AFB_1_) recognized as carcinogenic. Despite increasing global food safety awareness, there are limited data on the biodiversity of *Aspergillus* species, their mycotoxin-producing potential, and their phylogenetic relationships in dried meat sold in Ekiti State, Nigeria. This study seeks to investigate the diversity of *Aspergillus* species, their potential to produce mycotoxins, and their phylogenetic relationships in dried meat sold for human consumption in the region. The findings will help in implementing stricter regulations and raising awareness of the health risks associated with these fungi.

## 2. Materials and Methods

### 2.1. Study Area and Sampling Locations

This study was carried out at the department of Animal Health, Faculty of Natural Science and Agriculture, North-West University, South Africa, and samples were purchased from ten major local government areas of Ekiti State in Nigeria; these areas are as follows: Ado Ekiti and Igede town represent the central area, Ikole and Omuo town represent the eastern area, Aramoko and Ijero represent the western area, Oye and Otun represent the northern area, and Ilawe and Emure represent the southern area.

### 2.2. Sampling and Sample Preparation

One hundred and eight (108) samples of dry beef, each weighing about 200 g to 300 g, were obtained in triplicate from different markets in Ekiti State. The sampling design was a result of the availability of dry meat at the different markets. Representative samples were collected in plastic bags and sealed for transportation to the Food, Toxicology Research Laboratory, North-West University, Mafikeng Campus, South Africa, following all necessary protocols. All of the samples collected were meant for human consumption. On arrival at the laboratory, the meat samples were ground using a laboratory mill (IKA M20, Merck, Darmstadt, Germany) and stored at −20 °C for further analysis.

### 2.3. Mycological Screening of Samples

To determine the fungal population in the samples, the method described by [11], with slight modifications, was used. Potato Dextrose Agar (PDA) and Malt Extract Agar (MEA) were used. Potato Dextrose Agar and Malt Extract Agar were prepared by dissolving PDA (39 g) and MEA (39 g) in 1 litre of distilled water, respectively. Both agars were autoclaved at 121 °C and 15 psi for 15 min and cooled in a 50 °C water bath, and eight (8) millilitres (mL) each of 1% chloramphenicol and 1% streptomycin antibiotic solution were added to the agar before pouring into sterile Petri dishes.

### 2.4. Fungal Colony Counts

One (1) gram of each sample in triplicate was suspended in 9 mL of sterile Ringer’s solution vortexed for 2 min and serially diluted to make six dilutions (10^−6^) per sample. Ringer’s solution was prepared by dissolving two Ringer’s tablets in 1 litre of distilled water, which was autoclaved at 121 °C and 15 psi for 15 min. A 0.1 mL portion of the suspension was spread plated on PDA in triplicates and incubated at 28 °C for 48 h, and, subsequently, the colonies in each plate were counted, the average mean value was recorded as the fungal load for the 3 replicates in each sample, and the colony-forming unit (CFU/g) was calculated for each sample.
CFU/g = Number of colonies × reciprocal of the dilution factor
Plating volume (0.1 mL)

### 2.5. Fungal Isolation from Sample Cultures

One gram of each sample was suspended in 9 mL of sterile Ringer’s solution and vortexed for 2 min. A 0.1 mL aliquot of the suspension was spread plated in triplicate on Potato Dextrose Agar (PDA) and Malt Extract Agar (MEA) and incubated in the dark at 25 °C for 7 days. The individual fungal colony was carefully transferred into sterile solid MEA plates for final purification at 25 °C for 5 days before the fungal DNA was extracted. Identification of the isolated fungal was performed using a molecular method by PCR analysis and subsequently sequenced. The isolation occurrence of each species was calculated using the method described by [12].
$$\mathrm{Frequency}\;(\%)=\frac{\;Number\;of\;samples\;contaminated\;with\;a\;specie\;or\;genus}{Total\;number\;of\;samples}\times 100$$
$$\mathrm{Relative}\;\mathrm{Density}\;(\%)=\frac{Number\;of\;isolates\;of\;species\;or\;genus}{Total\;number\;of\;isolates}$$

### 2.6. Molecular Characterization of Fungal Isolates

#### 2.6.1. DNA Extraction

Deoxyribonucleotide (DNA) extraction was undertaken using a fungal/bacterial DNA extraction kit (Zymo Research Corporation, Tustin, CA, USA). Fungal spores from 4- to 5-day-old cultures were taken into the Bashing Bead^TM^ lysis tubes, and 750 μL of DNA lysis solution was added to the tubes. This was followed by lysing of the fungal cells using a disruptor genie bead beater fitted with a 2 mL tube holder assembly (Scientific Industries Inc., New York, NY, USA) and processed at maximum speed for 15 min and centrifuged at 10,000 rpm for 1 min. The supernatant was transferred to a zymo-spin IV spin filter in a collection tube and centrifuged at 10,000 rpm for 1 min. The filtrate was added to 1200 μL of fungal DNA-binding buffer and centrifuged in a Zymo-Spin IIIc column placed in a collection tube at 10,000× *g* for 1 min. This was followed by washing DNA in the column using a pre-wash buffer of 200 μL and a wash buffer of 500 μL. Finally, after washing, 100 μL of the DNA elution buffer was added to the column and centrifuged at 10,000× *g* for 30 min to elute the DNA. Then, the eluted DNA was stored at −70 °C for further analysis.

#### 2.6.2. Agarose Gel DNA Electrophoresis

Deoxyribonucleotide acid (DNA) gel electrophoresis was carried out by the preparation of 1% agarose gel 2 g of agarose (Fermentas Life Science, Vilnius, Lithuania) in 98 mL 1× TAE buffer (Fermentas Life Science, Lithuania), which was boiled in a microwave and cooled to approximately 60 °C. This was followed by the addition of 1 mL of ethidium bromide (Sigma-Aldrich, St. Louis, MO, USA) to the agarose solution. The agarose solution was poured into a casting chamber (Bio-Rad Laboratories, Hercules, CA, USA) and allowed to set. The casting chamber containing the solid agarose was further inserted into the electrophoresis tank (Bio-Rad Laboratories, CA, USA) filled with 1× TAE buffer. Fungal DNA (5 μL) was loaded into the wells. The chamber was closed and run at 400 V and 80 mA for 50 min, and DNA was viewed using the ChemDoc gel imaging system (Bio-Rad Laboratories, CA, USA).

#### 2.6.3. PCR Amplification of Extracted Genomic DNA

The amplification of the Internal Transcribed Spacer Region (ITS rDNA) of the fungal isolates from the dry meat was performed using Polymerase Chain Reaction (PCR) with the universal ITS 1 (TCCGTAGGTGAACCTGCGG) and ITS 4 region primers (TCCTCCGCTTATTGATATGC). These primers were obtained from Inqaba Biotechnical Industrial (Pty) Ltd. (Pretoria, South Africa). The PCR amplicons were analyzed through electrophoresis on 1% (*w*/*v*) agarose gel to check the anticipated size of the amplicons (670 bp) and viewed by using Chem Doc Image Analyzer [13]. The final PCR solution consisted of 12 μL of master mix, 1 μL each for forward and reverse primers, and 3 μL of DNA, constituted to a final volume of 25 μL with nuclease-free water. Polymerase chain reaction was performed using an Eppendorf 96-well Thermocycler (Eppendorf, Hauppauge, NY, USA), and cycling conditions were set as pre-dwelling at 94 °C for 1 min, 35 cycles of denaturation at 94 °C for 1 min, annealing at 54 °C for 1 sec, extension at 72 °C for 1 min 30 s, post-dwelling at 72 °C for 9 min, and holding at 4 °C until the samples were retrieved.

#### 2.6.4. Sequencing of PCR Products

The purified PCR products were sequenced at Inqaba Biotechnical Industrial (Pty) Ltd., Pretoria, South Africa, using the PRISM^TM^ Ready Reaction Dye Terminator Cycle Sequencing Kit with the dideoxy chain cessation method and electrophoresed with a model ABI PRISM^®^ 3500XL DNA Sequencer (Applied Biosystems, Foster City, CA, USA) by means of following the manufacturer’s instructions.

#### 2.6.5. Sequence Analysis

Finch TV software, version 1.4.0, was employed for the analysis of chromatograms, (sense and antisense) ensuing from sequencing reactions for superior sequence assertion. The subsequent chromatographs were edited using Bio Edit Sequence Alignment Editor according to [14], after which the resultant consensus ITS rDNA sequences were imported into the NCBI database for blasting using the Basic Local Alignment Search Tool (BLAST) for homology in the identification of the likely organisms [15]. The sequences were subsequently deposited in GenBank for the allocation of accession numbers.

#### 2.6.6. Detection of Aflatoxin-Producing Genes of Aspergilla

The DNA of assumed *Aspergilla* was analyzed for the presence of five important aflatoxin-producing genes (*aflR*, *aflJ*, *aflM*, *aflD*, and *omt-A*) in the isolated *Aspergillus flavus*, which was detected using a previously used and reported set of primers by [16], using primers described in Table 1. The primers were provided by Inqaba Biotechnical Industrial (Pty) Ltd., Pretoria, South Africa, and the PCR was conducted using a PCR Thermal Cycler (Applied Biosystems).

#### 2.6.7. Optimization of Polymerase Chain Reaction

Polymerase Chain Reaction conditions were optimized separately for the target genes. A reaction volume of 25 μL, containing 8.5 μL nuclease-free water, 12.5 μL PCR Master Mix, 1 μL of oligonucleotide forward and reverse primers (10 μm), and 2 μL template DNA mixed in the PCR tubes, were used [21]. The thermal cycle conditions are also shown in Table 1, with varying annealing temperatures ranging from 58 to 75 °C for the five genes. The PCR-amplified products were checked on 1% gel by electrophoresis and visualized under the gel documentation system for electrophoretic bands at the various base pair regions for each gene.

### 2.7. Phylogenetic Analysis

The evolutionary study was carried out using Molecular Evolutionary Genetics Analysis (MEGA) 5 [22], and the neighbour-joining method was used for evolutionary history, according to [23]. The evolutionary distances were calculated using the parameter method of [24], using the transversional substitutions per site unit of the number.

### 2.8. Statistical Analyses

Statistical analyses were performed using Microsoft Excel 2013 for Windows 2010. Descriptive statistics were employed to present the data collected from individual species isolated from each dried meat sampled and was considered one isolate. The concentrations were calculated based on the average recovery values acquired for each analyte. This was carried out using IBM SPSS version 25 software, with a statistical significance level of *p* < 0.05.

## 3. Results and Discussion

### 3.1. The Fungal Counts and Distribution of Fungal Isolates in Dry Meat

A total of 118 fungal isolates were isolated through the ITS section amplification and sequencing of the PCR amplicons of the isolates. One hundred percent (100%) of the dry meat analyzed was contaminated with *Aspergillus* of different species, with fungal counts ranging from 1.0 to 9.5 log^10^ cfu/g with a mean of 3.1 log^10^ cfu/g (Table 2). The dried meat sample from Ikole had the highest colony-forming unit range of 2.0 × 10^2^–1.6 × 10^4^, with an average mean value of 2.6 × 10^3^, though with 10 different types of fungi species contaminating them. The least colony-forming unit range was recorded in Ilawe meat samples with a range value of 1.0 × 10^1^–4.5 × 10^2^, with an average mean value of 1.3 × 10^2^, and a total number of 11 fungi species contaminating them. The low fungal counts of the dried meat might be due to the low moisture content, which reduced the growth of fungi in the dry meat [25]. A similar report of fungal count (1.8 × 10^3^–4.6 × 10^3^ cfu/g) was reported for sun-dried meat samples (Kundi) obtained from major markets in Ibadan, Oyo State, in Nigeria [26].

This result shows that the majority of the fungi species isolated from the different locations are peculiar to the sampling environment and have been previously reported from other food samples, which implies that their occurrence in dried meat samples could be due to cross-contamination, as many of the dried meats are seen displayed close to other agricultural commodities (Figure 1). In addition, the fungi biodiversity could also be due to storage, the selling environment, and possible interactions with buyers. Customers are known to touch dried meats when buying them in order to select the ones that are deemed fit or of good quality (Figure 1). Another likely means of contamination is cross-contamination from the seller, who frequently touches the dried meat either when helping the buyer to select some good meat devoid of visible fungi contamination, when displaying the dried meat for sale, when packing after the end of daily sales, or when giving change to the buyers (Figure 1). Additionally, vehicular and human movements can also contribute to meat contamination as it moves from one end to the other. This movement sometimes raises dust, which eventually perches on or contaminates the dried meat, as they are not covered or protected from the environment where they are sold.

The overall fungal contamination results (Figure 2) showed that all the sampled locations have *Aspergillus* and *Rhizopus* fungal genera. These were found to contaminate the dry meat across the locations, whereas *Penicillium* genera were found in 6 out of the 10 locations investigated. The remaining fungal genera were found randomly in dry meat samples from other locations, with at least four different types occurring in Ikole, Oye, and Igede, respectively, whereas the highest fungal genera with 8 fungi were recorded in Ado and Omuo, followed by Aramoko with 7 fungal genera contamination. The dominance of *Aspergillus* and *Rhizopus* species in dry meat samples analyzed corresponds with the high incidence of *Aspergillus* and *Rhizopus* species in regions with high temperatures and humidity [27], with regard to the climatic conditions of Ekiti State (Nigeria). The state enjoys a tropical climate with two distinct seasons. These are the rainy season (April–October) and the dry season (November–March). The temperature ranges between 21 °C and 28 °C with high humidity. This finding is in agreement with the known knowledge of *Aspergillus* species as generally isolated from soil, air, enclosed surroundings, and agricultural products [28,29]. *Fusarium* and *Cladosporium* species were also seen to contaminate the bulk of the sampled meats. *Penicillium* has been reported to infect maize and maize-based produce in West Africa at diverse yield periods [30]. Additional species found to occur with a lesser prevalence comprised *A. aeneus*, *A. alabamensis*, *A. amstelodami*, *A. awamori*, *A. chevalieri*, *A. melleus*, *A. ruber*, *A. tubingensis*, *A. unguis*, *Byssochlamys spectabilis*, *Chaetomium erectum*, *Cochliobolus* sp., *Epicoccum sorghinum*, *Fusarium equiseti*, *Fusarium incarnatum*, *Paecilomyces formosus*, *Penicillium funiculosum*, *Penicillium mallochii*, *Periconia macrospinosa*, *Phoma herbarum*, and *Talaromyces funiculosus.* The higher incidence of *Aspergillus*, *Rhizopus*, and *Penicillium* species in dry meat samples collected across the state could be because *Aspergillus*, *Rhizopus*, and *Penicillium* species are known to be storage fungi that usually contaminate food products under storage [31,32,33]. The occurrence of these fungal species in dry meat is a pointer to possible mycotoxin contamination because the majority of them are known producers of main mycotoxins [34]. For instance, *A. flavus* and *A. niger* are known producers of aflatoxins in agricultural produce [35,36].

### 3.2. The Occurrence of Aspergillus Species in Dried Meat across Different Locations

The occurrence of *Aspergillus* specie*s* differs from each location as shown in Figure 3. The findings showed that *Aspergillus flavus* was not found in Oye and Ise/Emure; *Aspergillus oryzae* was not found in Aramoko, Oye, Otun, Omuo, and Igede; *Aspergillus aeneu* was found in Aramoko, and *Aspergillus alabamensis* was found in Ijero; *Aspergillus awamori*, *Aspergillus chevalieri*, and *Aspergillus tubingensis* were found in Ado; *Aspergillus unguis* was found only in Oye; and *Aspergillus bombycis* was found in Ilawe, Aramoko, and Omuo. *Aspergillus intermedius* was found in Ijero, Ado, and Omuo.

The possession of the aflatoxin gene is shown in Figure 4, where M: 1 kb DNA ladder; lane 1, 3, 4, 7, 10, 11, 12, 16, 17 and 18 shown fungal gene expression while lane 19 is the negative controls. The b gel electrophoretic pattern is shown for PCR products expressing the *aflR* gene at the 1000 bp region. M: 1 kb DNA ladder; lanes 1, 3, 10, 12, 15, and 18: positive isolates, lane 19: negative control, and lanes 2, 4–9, 11, 13, 14, 16, and 17: negative isolates. The c gel electrophoretic arrangement is shown for PCR products expressing the *aflD* gene (Nor-1) at the 400 bp region. M: 1 kb DNA ladder; lane 19: negative control, and lanes 1–18: amplification of the *nor* gene in positive isolates. The d gel electrophoretic pattern for PCR products expressing the *omt-A* gene at 797 bp is shown. M: 1 kb DNA ladder; lanes 1, 2, 4, 5, 10, 12, 14, 17, and 18: positive isolates, lane 19: negative control; and lanes 3, 6–9, 11, 13, –15, and 16: amplification of *omt-A* gene in positive isolates. The e gel electrophoretic pattern for PCR products expressing the *aflM* gene (Ver) at 600 bp is shown. M: 1 kb DNA ladder; lanes 4, 5, 6, 8,13, and 15: amplification not at the anticipated band size; lane 19: negative control; and lanes 1,3,16, and 18: positive isolates. The f Gel electrophoretic pattern for PCR products expressing the *aflJ* gene at the 684 bp region is shown. M: 1 kb DNA ladder; lanes 1, 3, 4, 8, 9, 11, and 17: positive isolates; lane 19: negative control; and lanes 3, 18, and 24: negative isolates.

This study found that 75% of the isolated *Aspergillus flavus* strains contained both the aflatoxin biosynthetic regulatory genes, *aflR* and *aflJ*, along with three key genes in the aflatoxin biosynthetic pathway: *omt-A*, *aflD (nor-1*), and *aflM (ver-1)* (Table 2). Other isolates showed the presence of one or more of these genes but not the full set. The detection of these genes is concerning, as it indicates the potential for mycotoxin production if environmental conditions become favourable for fungal growth. It has been reported that *aflR*, *omt-A*, *aflD*, and *aflM* remain the most vital genes from all the 25 reported genes in the aflatoxin biosynthetic group that control aflatoxin production [37]. The *aflR* gene controls the initiation of the transcript of pathway genes and aflatoxin biosynthesis [38], but its occurrence in the isolates may not automatically result in aflatoxin production since the aflatoxin pathway is controlled by several mechanisms [39,40]. Also, *aflJ* is likewise an aflatoxin biosynthetic controlling gene identical to the *aflR*; the dual genes are different transcripts with separate sponsors. It is essential for the expression of other genes in the aflatoxin group and is also involved in the transformation of pathway transitional products to aflatoxins [41]. The *aflR* protein can bind the promoter area of individual aflatoxin synthesis genes and trigger gene expression [42]. Also, aflatoxin biosynthesis is controlled by many ecological factors, for example, light [43], carbon source, temperature, and pH [44]. Table 3 gives overall information about the sequence of *A. flavus* that shows gene expression to aflatoxin biosynthetic genes among all the *Aspergillus* species isolated from the dried beef meat as submitted to GenBank.

The maximum likelihood phylogenetic tree based on the partial ITS rDNA gene sequence, showing the phylogenetic relationships between identified *Aspergillus* species and the most closely related *Aspergillus* strains from GenBank, is shown in Figure 5. The numbers at the nodes indicate the levels of bootstrap support based on 1000 resampled data sets. Only values greater than 50% are shown. The scale bar indicates 0.5 nucleotide substitution per site. *Aphanophora eugeniea* is set as the outgroup. Sequences obtained in this study are denoted with a red circle. This result is divided into six main clades, comprising the following. Clade I: *Aspergillus unguis* and *A. aeneus* shared 98% and 99% with the reference of the same species from GenBank. Clade II: *Aspergillus fumigatus* shared 97%. *A. awamori*, *A. niger* (85%), and *A. tubingensis* shared 82%. Clade III: *A. ustus* shared 68% with *A. ustus* in GenBank. *A. cristatus* shared 68% homology with *A. ustus* and 99% with *A. candidus*. *A. candidus* also shared 68% with the reference strain of *A. candidus* from GenBank. This similarity index is quite low, except for *A. ustus*, since it is below the 70% anticipated borderline for the degree of similarity according to [45]. *A. ruber* shared 98%, with *A. amstelodami* as the closest relative. The samples of *A. cristatus*, *A. intermedius*, *A. amstelodami*, and *A. chevalieri* shared 81% with the reference strain from GenBank. Clade IV *A. alabamensis* shared 79% with the reference strain, while Clade V *A. oryzae*, *A. flavus*, and *A. bombycis* shared 95% with the reference strain. Clade VI *A. tamarii* shared 98% with the reference strain. *A. ochraceus* shared 99% with the *A*. *ochraceus* and *A. melleus* reference strains. This high homology percentage shows that they possessed nearly the same nucleotide signature as their relative counterpart from GenBank [46]. This high similarity index expressed by these fungi is beyond the 70% borderline of the expected degree of relatedness [45]. *A. melleus* from this study shared 60% with *A. ochraceus* and 68% with *A. melleus* reference from GenBank. This fungus has a different nucleotide signature, and its low threshold makes it prone to being wiped out [47]. This could be a novel *Aspergillus* species.

## 4. Conclusions

Based on the findings of this study, *Aspergillus* species isolated from dried meat in Ekiti State, Nigeria, showed significant biodiversity and a high potential for mycotoxin production. The presence of key aflatoxin biosynthetic genes in a majority of the isolates highlights the potential health risks associated with consuming contaminated dried meat. Given these findings, there is a need for stricter regulations and increased awareness of the dangers posed by *Aspergillus* contamination to ensure food safety and protect public health.

## Figures and Tables

**Figure 1 foods-13-03221-f001:**
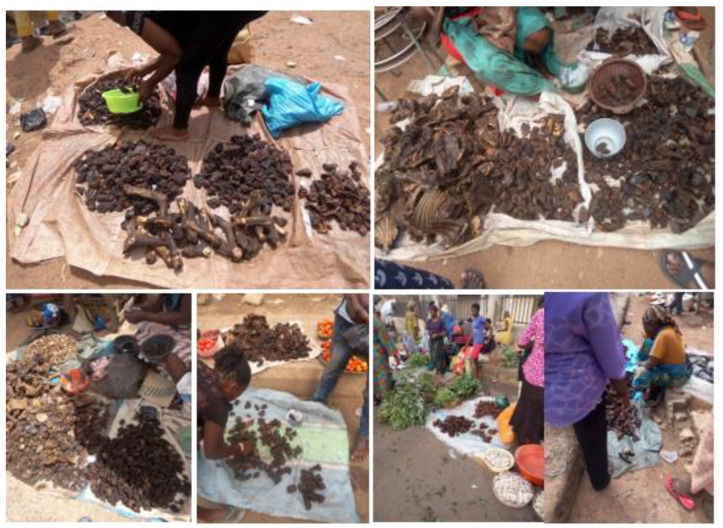
Open display methods of selling dried meat across the sampling locations in Ekiti State.

**Figure 2 foods-13-03221-f002:**
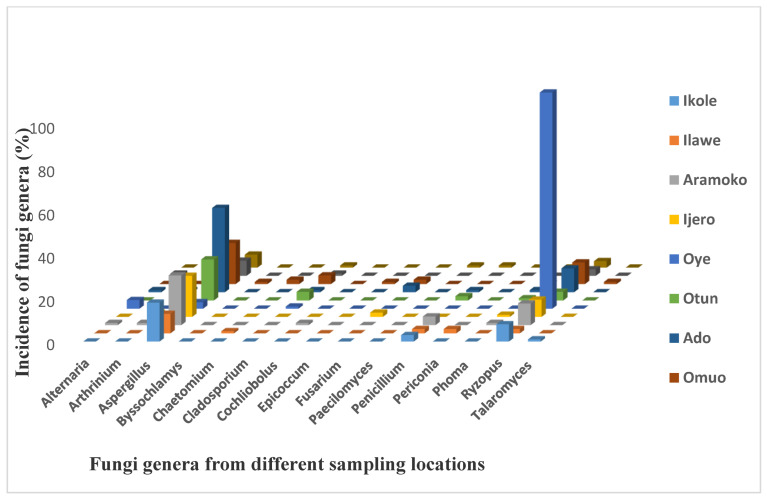
Incidence (%) of different fungi genera contaminating dry meat according to various locations.

**Figure 3 foods-13-03221-f003:**
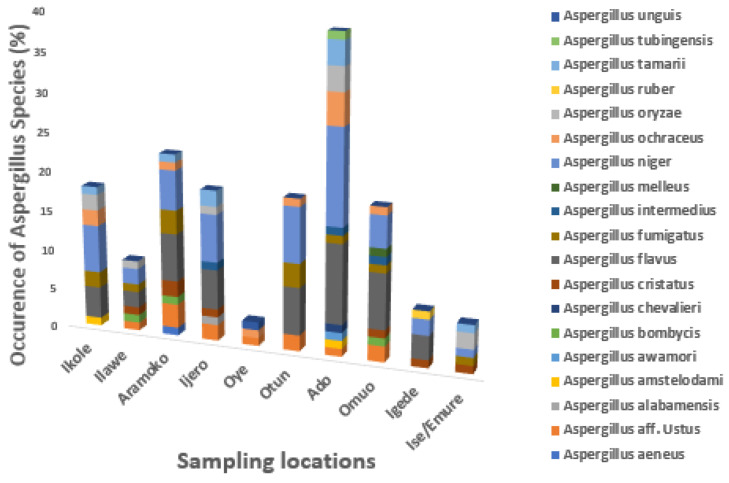
The occurrence of *Aspergillus* species in dried meat across different locations.

**Figure 4 foods-13-03221-f004:**
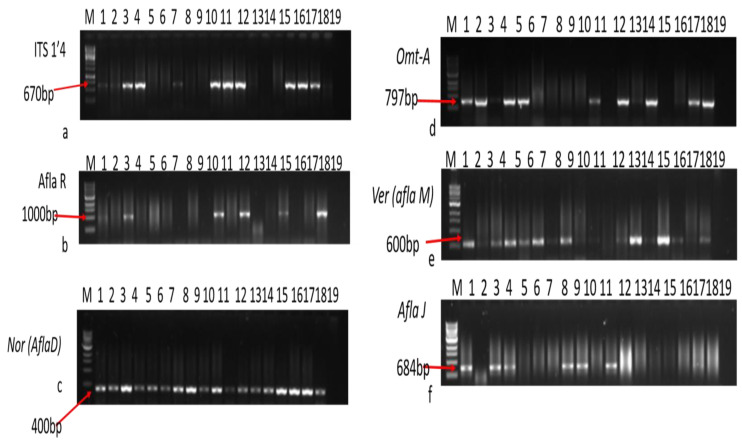
Gel electrophoretic analysis for PCR products showing *A. flavus* gene presence at different bp regions.

**Figure 5 foods-13-03221-f005:**
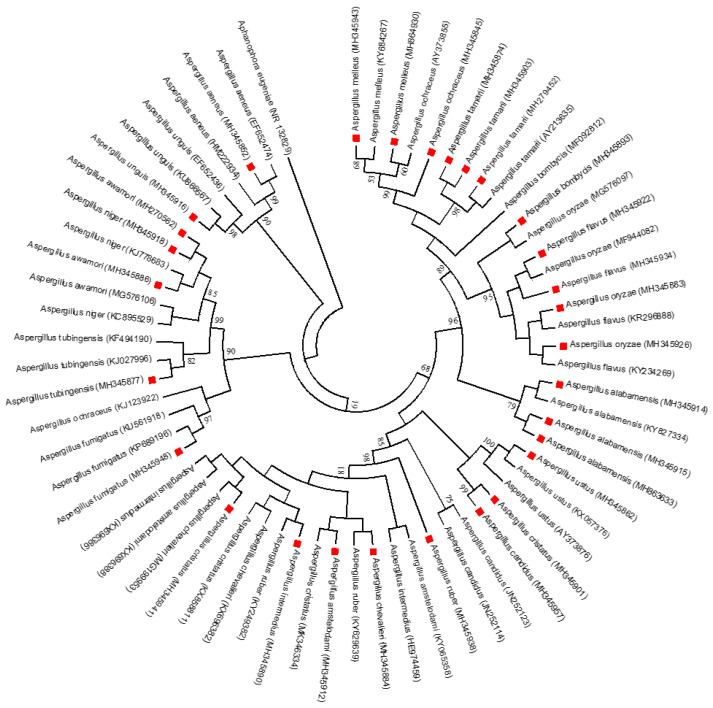
Maximum likelihood phylogenetic tree for identified *Aspergillus* species and their most closely related strains from GenBank.

**Table 1 foods-13-03221-t001:** Sequences of the nucleotide primers used in this study.

Primer Code	Target Gene	Primer Sequences 5′–3′	Product Size (bp)
Nor-F	*aflD* (*nor-1*)	ACCGCTACGCCGGCACTCTCGGCAC	400
Nor-R		GTTGGCCGCCAGCTTCGACACTCCG	
Ver-F	*aflM* (*ver-1*)	GCCGCAGGCCGCGGAGAAAGTGGT	600
Ver-R		GGGGATATACTCCCGCGACACAGCC	
Omt-F	*aflP* (*omtA*)	GTGGACGGACCTAGTCCGACATCAC	797
Omt-R		GTCGGCGCCACGCACTGGGTTGGGG	
AflR-for	*AflR*	TATCTCCCCCCGGGCATCTCCCGG	1000
AflR-rev		CCGTCAGACAGCCACTGGACACGG	
AflJ-for	*aflS* (*aflJ*)	TGAATCCGTACCCTTTGAGG	684
AflJ-rev		GGAATGGGATGGAGATGAGA	

References: [17,18,19,20].

**Table 2 foods-13-03221-t002:** Colony-forming unit (Cfu/g) of fungal species recovered from dried meat per location.

Location	CFU g^−1^ Range	Mean	No of Fungi Isolates
Ikole	2.0 × 10^2^–1.6 × 10^4^	2.6 × 10^3^	10
Ilawe	1.0 × 10^1^–4.5 × 10^2^	1.3 × 10^2^	11
Aramoko	1.0 × 10^2^–6.9 × 10^3^	1.5 × 10^3^	16
Ijero	2.0 × 10^2^–3.2 × 10^2^	1.5 × 10^2^	13
Oye	5.0 × 10^1^–4.1 × 10^2^	2.9 × 10^2^	6
Otun	1.0 × 10^1^–1.9 × 10^2^	5.4 × 10^1^	11
Ado	2.0 × 10^1^–2.1 × 10^3^	3.2 × 10^2^	19
Omuo	3.0 × 10^1^–6.4 × 10^2^	1.6 × 10^2^	16
Igede	9.0 × 10^1^–3.2 × 10^2^	1.0 × 10^2^	7
Ise/Emure	1.0 × 10^1^–2.5 × 10^2^	9.5 × 10^1^	9

**Table 3 foods-13-03221-t003:** The potential of isolated *A. flavus* to produce aflatoxin.

S/No	Name of Isolates	Nor *afLD*	Ver *afLM*	Omt *afLP*	afLR	afLJ	AP	Accession Number	NCBI % Similarity
1	*A. flavus*	+	+	+	+	+	P	MH345936	99
2	*A. flavus*	+	+	+	+	+	P	MH345870	99
3	*A. flavus*	+	+	+	+	+	P	MH345846	100
4	*A. flavus*	+	+	+	+	+	P	MH345934	100
5	*A. flavus*	+	+	+	+	+	P	MH345922	100
6	*A. flavus*	+	+	+	+	+	P	MH345895	99
7	*A. flavus*	+	+	+	+	+	P	MH345879	99
8	*A. flavus*	+	+	+	+	+	P	MH345959	99
9	*A. flavus*	+	+	+	+	+	P	MH345842	99
10	*A. flavus*	+	+	+	+	+	P	MH345932	99
11	*A. flavus*	+	+	+	+	+	P	MH345881	100
12	*A. flavus*	+	+	+	+	+	P	MH345952	100
13	*A. flavus*	+	+	+	+	+	P	MH345913	99
14	*A. flavus*	+	+	+	+	+	P	MH345910	100
15	*A. flavus*	+	+	+	+	+	P	MH345946	99

P = potential to produce aflatoxin, + means positive gene expression to the aflatoxin genes, AP = aflatoxin production.

## Data Availability

The datasets backing the conclusion of this article are included within the article, and the nucleotide sequences presented in this article are accessible through GenBank.

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
