# Peer review of "Biodiversity of Aspergillus Species and Their Mycotoxin Production Potential in Dry Meat"

_foods, 2024, doi:10.3390/foods13203221_

Round 1

Reviewer 1 Report

Comments and Suggestions for Authors

This is an interesting work, investigating Aspergillus species diversity in dried meat products. The results highlighted a great safety risk with the dried meat products. The manuscript is generally well-organized, and all the data, tables, figures are sufficient. I have only one minor suggestion, where the content of  mycotoxin in contaminated products should also be accurately quantified, so as to cross-confirm the previous findings. 

Author Response

This is an interesting work, investigating Aspergillus species diversity in dried meat products. The results highlighted a great safety risk with the dried meat products. The manuscript is generally well organized, and all the data, tables, and figures are sufficient. I have only one minor suggestion: the content of mycotoxin in contaminated products should also be accurately quantified so as to cross-confirm the previous findings. 

Respose: The reviewer's minor suggestion to accurately quantify the mycotoxin content in the contaminated product was done in our previous publication titled 'Dada, T. A., Ekwomadu, T. I., & Mwanza, M. (2020). Multi mycotoxin determination in dried beef using liquid chromatography coupled with triple quadrupole mass spectrometry (LC-MS/MS). Toxins, 12(6), 357. 

His suggestions on 'can be improved' if the results are clearly presented, and the conclusion supported by the results will be properly considered as it also appears in the second reviewer's comment.

Thanks for the positive comments and for your time to review our work

Reviewer 2 Report

Comments and Suggestions for Authors

The manuscript concerns the occurrence of mycotoxin producing fungi in dried meat. The paper  has some flaws that need to be corrected. The details are listed below:

L2: Latin names in italics. Correct throughout the paper

L16: indicate the percentage in brackets and remove the names of fungi in L17

L20: ‘with gene expression for aflatoxin’ – rephrase; ‘the other three’ – remove

L66-70: the study is based on mycotoxins potential in dry meat, but mycotoxins are superficially included in the Introduction. Describe the ubiquitous evidence of mycotoxins in different food products, their health effects and health risk. Refer to https://doi.org/10.1016/j.foodchem.2024.140222

L72, 74: Aspergillus with uppercase and italics

L75: actually mycotoxin contamination was not investigated in this study

L100: Potato Dextrose Agar

L311: it is the presence of the genes, but not their expression

L329-335: why these genes are present only in selected isolates – discuss

L334-335: do not use the term expression to the presence of these genes in samples

L384: do not repeat the results in the conclusions. Summarize the main results

Comments on the Quality of English Language

Minor editing of English language required.

Author Response

For research article Biodiversity of Aspergillus species and their mycotoxin production potentials in dry meat.

Response to Reviewer 2 Comments

1. Summary

2. Questions for General Evaluation

Reviewer’s Evaluation

Response and Revisions

Does the introduction provide sufficient background and include all relevant references?

Must be improved

The introduction has been improved, page 2, lines 60-76 in the revised manuscript

Are all the cited references relevant to the research?

Not applicable

Not applicable

Is the research design appropriate?

Can be improved

We believe that this research design is appropriate. This is an established design which has been used by other researches and duly referenced

Are the methods adequately described?

Must be improved

The method used are well-known methods that have been widely used. So, we believe that they were adequately described and properly referenced.

Are the results clearly presented?

Must be improved

The result has been clearly re-presented where we believe clarity was required as there was no specific result mentioned for improvement, These changes can be found in page 9, line 331-336

Are the conclusions supported by the results?

Must be improved

This conclusion has been improved, concisely summarized, and rephrased to reflect the result. Page 11, line 386-392

3. Point-by-point response to Comments and Suggestions for Authors

Comments 1: [L2: Latin names in italics.]. Correct throughout the paper]

Response 1: [Latin names in italics has been corrected in L2 page 1 and also corrected throughout the paper] Thank you for pointing this out. I/We agree with this comment. Therefore, I/we have done the correction [this changed has been done and effected in the manuscript Aspergillus and can be found in L2, page 1]

Comments 2: [L16: indicate the percentage in brackets and remove the names of fungi in L17]

Response 2: Agree. I/We have modified it to indicate the percentage and removed the fungi names as suggested in L17. This change can be found in the revised manuscript on page 1, abstract, L17]

Comments 3: [L20: ‘with gene expression for aflatoxin’ – rephrase; ‘the other three’ – remove]

Response 3: Agree. I/We have rephrased the statement to reflect Aspergillus flavus accounted for (24.7%) of all the Aspergillus species isolated with the presence of gene needed for aflatoxin production. While the ‘other three’ has been removed and suggested. This change can be found in the revised manuscript on page 1, abstract, L19 & 20]

Comments 4: [L66-70: the study is based on mycotoxins potential in dry meat, but mycotoxins are superficially included in the Introduction. Describe the ubiquitous evidence of mycotoxins in different food products, their health effects and health risk. Refer to https://doi.org/10.1016/j.foodchem.2024.140222]

Response 4: Agree. I/We have rephrased the entire statement to reflect Several studies have shown the presence of fungi and mycotoxins in kundi, fish, banda, and other dried products. Some of them are shown to have the potential to produce secondary metabolites known as mycotoxins, which are capable of remaining in the product long after the fungal has died [9,10].

Consumption of meat contaminated with Aspergillus species can expose consumers to aflatoxicosis, a disease caused by the ingestion of aflatoxins. These mycotoxins, mainly produced by Aspergillus flavus and Aspergillus parasiticus, are among the most toxic, with Aflatoxin B1 (AFB1) recognized as carcinogenic. Despite increasing global food safety awareness, there is limited data on the biodiversity of Aspergillus species, their mycotoxin-producing potential, and their phylogenetic relationships in dried meats sold in Ekiti State, Nigeria. This study seeks to investigate the diversity of Aspergillus species, their potential to produce mycotoxins, and their phylogenetic relationships in dried meat sold for human consumption in the region. The findings will help in implementing stricter regulations and raising awareness of the health risks associated with these fungi. This change can be found in the revised manuscript on page 2, paragraphs 4 & 5, L60-76]

Also, the health effect has been added to the mycotoxin introduction in Line 63-66, line 60 & 61 already talked about the relevant food products and others products with mycotoxin contamination.

Comments 5: [L72, 74: Aspergillus with uppercase and italics]

Response 5: Agree. I/We have rephrased the word to reflect Aspergillus with initial uppercase and italics in paragraph 5, page 2, line 67, 69, 71 & 73 in the revised manuscript.

Comments 6: [L75: actually, mycotoxin contamination was not investigated in this study]

Response 6: Agree. I/We have rephrased the entire sentence to reflect This study seeks to investigate the diversity of Aspergillus species, their potential to produce mycotoxins, and their phylogenetic relationships in dried meat sold for human consumption in the region. This change can be found in the revised manuscript on page 2, paragraph 5, L73-75]

Comments 7: [L100: Potato Dextrose Agar]

Response 7: Agree. I/We have changed the word to reflect initial capital and corrected to be Potato Dextrose Agar. This change can be found in the revised manuscript on page 2, L98 & 99]

Comments 8: [L311: it is the presence of the genes, but not their expression]

Response 8: Agree. I/We have changed the word to reflect the presence and deleted expression. This change can be found in the revised manuscript on page 9, L 313]

Comments 9: [L329-335: why these genes are present only in selected isolates – discuss]

Response 9: Agree. I/We have rephrased the sentence to convey the report/explanation we are trying to put forward. This change can be found in the revised manuscript on page 9, paragraph 2, L330 - 335]

Comments 10: (L334-335: do not use the term expression to the presence of these genes in samples)

Response 10: This term has been taken care of in the process of rephrasing the entire statement in Comment 9 above.

Comments 11: (L384: do not repeat the results in the conclusions. Summarize the main results)

Response 11: This term has been taken care of in the process of rephrasing the entire statement in Comment 9 above.

Response to Comments on the Quality of English Language

Response 1: (The entire manuscript has been re-read with the corrections done where necessary.)

Additional clarifications

The corrections carried out as suggested by the reviewers are marked with red in the revised manuscript.
